# FOUNDATION MODELS SECRETLY UNDERSTAND NEURAL NETWORK WEIGHTS: ENHANCING HYPERNETWORK ARCHITECTURES WITH FOUNDATION MODELS

**Jeffrey Gu, Serena Yeung-Levy**
Institute for Computational and Mathematical Engineering (ICME), Department of Biomedical Data Science
Stanford University
Stanford, CA 94305, USA
{jeffgu, syyeung}@stanford.edu

## ABSTRACT

Large pre-trained models, or foundation models, have shown impressive performance when adapted to a variety of downstream tasks, often out-performing specialized models. Hypernetworks, neural networks that generate some or all of the parameters of another neural network, have become an increasingly important technique for conditioning and generalizing implicit neural representations (INRs), which represent signals or objects such as audio or 3D shapes using a neural network. However, despite the potential benefits of incorporating foundation models in hypernetwork methods, this research direction has not been investigated, likely due to the dissimilarity of the weight generation task with other visual tasks. To address this gap, we (1) show how foundation models can improve hypernetworks with Transformer-based architectures, (2) provide an empirical analysis of the benefits of foundation models for hypernetworks through the lens of the generalizable INR task, showing that leveraging foundation models improves performance, generalizability, and data efficiency across a variety of algorithms and modalities. We also provide further analysis in examining the design space of foundation model-based hypernetworks, including examining the choice of foundation models, algorithms, and the effect of scaling foundation models.

## 1 INTRODUCTION

Foundation models, models that are pre-trained using self-supervision on diverse large-scale datasets and are readily adaptable to a wide variety of downstream tasks (Bommasani et al., 2021), have revolutionized AI as these models have formed the backbone for state-of-the-art models in many tasks across a wide range of modalities, such as zero-shot image classification (Radford et al., 2021) and segmentation (Kirillov et al., 2023). However, the use of foundation models has not been investigated for many downstream tasks where they may be useful.

Hypernetworks, which are neural networks that produce or adapt some or all of the weights of another neural network, have been investigated as a way to create adaptive layers (Ha et al., 2016; Ba et al., 2016; Goyal et al., 2019), perform neural architecture search (Brock et al., 2017; Zhang et al., 2018a), meta-learning (Andrychowicz et al., 2016; Zhao et al., 2020) and multi-task learning (Tay et al., 2020), continual learning (Von Oswald et al., 2019), and more. One major area of hypernetwork research is using hypernetworks as a means of conditioning or creating generalizable implicit neural representations (INRs) (Mescheder et al., 2019; Sitzmann et al., 2019; 2020b; Chen & Wang, 2022; Gu et al., 2023; Kim et al., 2023; Lee et al., 2024). INRs, also known as coordinate-based neural networks or neural fields, represent signals or objects using a neural network and have emerged as a continuous and memory-efficient alternative to traditional discrete representations (Sitzmann et al., 2020b). Typically, INRs are trained to represent a single object from many partial sensor observations of that object. Generalizable INRs improve this training framework by leveraging additional data to improve INR quality (Tancik et al., 2021; Chen & Wang, 2022; Gu et al., 2023; Kim et al., 2023; Lee et al., 2024), training efficiency (Tancik et al., 2021), and speed (Hong et al., 2023) as well as allow

the generation of INRs for objects that would otherwise have insufficient partial observations (Hong et al., 2023).

Despite the success of foundation models for other tasks, there has been little investigation into adapting foundation models to improve hypernetworks and generalizable INRs, as none of the state-of-the-art methods (Chen & Wang, 2022; Gu et al., 2023; Kim et al., 2023; Lee et al., 2024) leverage foundation models. We believe that this is due to modality gap between neural network weights, the output of the hypernetwork task, which differs significantly from the outputs of more typical tasks such as image classification. We address this gap in the existing literature by answering the following questions: Do foundation models improve hypernetwork performance on the generalizable INR task? In which ways do they improve hypernetworks? And if they do, how should one design a hypernetwork from a foundation model? To answer these questions, we first augment Transformer-based generalizable INR architectures (Chen & Wang, 2022; Gu et al., 2023; Kim et al., 2023) with foundation models and show that adaptation via fine-tuning improves downstream task performance, generalization to unseen classes, and data efficiency. We also show that the foundation model approach can outperform existing approaches even when the foundation model features are frozen and only linear heads and extra tokens are trained on top of these frozen features to produce the weights of each layer. In addition, we provide an analysis of the design space of adapting foundation models to hypernetworks through targeted experiments exploring 1) the choice of foundation model, 2) the choice of algorithm, and 3) scaling properties. Finally, we show that the performance is robust by examining different modalities.

The contributions of our paper are as follows: first, using a framework based on existing Transformer-based hypernetworks, we show that foundation models improve hypernetwork performance on the generalizable INR task on different modalities. Second, we perform additional experiments to analyze many different facets of performance, examining generalization to unseen data, data efficiency, and parameter efficient approaches. We also provide additional analysis on the design space of adapting foundation models to hypernetworks, the choice of algorithm, examining the choice of foundation model, how performance scales with the number of foundation model parameters.

## 2 BACKGROUND AND SETUP

In this section, we first describe how hypernetworks can be used to generate generalizable INRs. We then describe the architecture of the foundation model-based hypernetwork we use, which is closely based on the Trans-INR (Chen & Wang, 2022), a Transformer-based hypernetwork architecture that produces an INR in one forward pass of the hypernetwork. We then discuss the design space of hypernetworks and describe how parameter-efficient fine-tuning can be done using prompt tuning (Jia et al., 2022).

**Hypernetworks for INRs** Define a signal $I : X \rightarrow Y$ as a function that maps coordinates $X$ to a space of quantities $Y$. An implicit neural representation (INR) represents the signal $I$ by parameterizing it with a neural network $f_\theta$ with weights $\theta$: $I(x) \approx f_\theta(x), \forall x$. INRs are typically trained with only partial observations $v$ of the signal $I$ using a forward map $F$ that maps the outputs $y$ to the partial observations $v'$, and are supervised with a reconstruction loss between $v, v'$. For example, in Neural Radiance Fields (NeRF) (Mildenhall et al., 2021) the signal parameterized is the radiance field of a 3D scene ($I$) and the partial observations $v$ are 2D views of the scene. The forward map $F$ is volume rendering, an operation which takes the radiance field and produces the predicted partial 2D view $v'$. The INR is then supervised with a reconstruction loss between $v$ and $v'$, which is the mean-squared error (MSE) in this example.

A hypernetwork is a neural network that produces the weights of another neural network. In our case, a hypernetwork $g_\phi$ produces the weights of an INR $f_\theta$ given some partial observations $v$ of the signal $I$: $g_\phi(v) = \theta$. The hypernetwork model is then supervised by a reconstruction loss between the $v$ and the predicted partial observation $v' = F(f_{g_\phi(v)}(x))$, where $x$ are the coordinates corresponding to $v$ and $F$ is the forward map. Only the hyperparameter weights $\phi$ are optimized using backpropagation.

**Foundation Model Framework for Hypernetworks** We base the foundation model framework for analysis (Figure 1) on the Trans-INR architecture (Chen & Wang, 2022), a Transformer-based hypernetwork architecture for generalizable INR. It consists of four main components: (1) a pre-

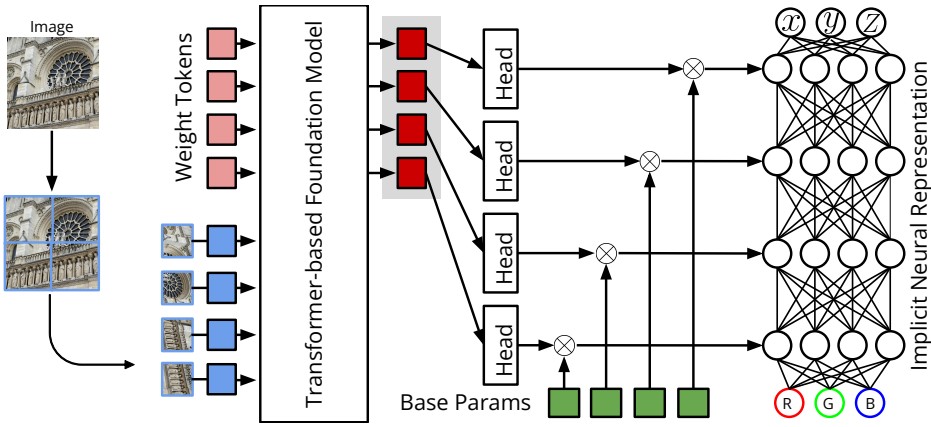

Figure 1: An overview of the hypernetwork-foundation model framework. First, an image is tokenized and concatenated with learnable weight tokens. Second, all tokens are encoded by a pre-trained foundation model encoder (Eq. 1). Tokens are then grouped, transformed using linear heads $\texttt{Head}_k$, and multiplied element-wise $\otimes$ with the base parameter $\texttt{BaseParam}_k$. (Eq. 2), and normalized (not shown). The resulting masked weights are then used to instantiate an implicit neural representation (INR). The INR can then be trained as usual.

trained Transformer foundation model consisting of an embedding layer $\texttt{Embed}$ and $d$-dimensional Transformer encoder $\texttt{Enc}$ consisting of attention blocks $\{B_i\}_{i=1}^N$, (2) extra learnable input tokens $\{w_j^0\}_{j=1}^q$, (3) an INR $f$, generally a ReLU MLP with positional encoding composed of layers $L_k$, and (4) a learnable linear head $\texttt{Head}_k$ for each layer $L_i$ of the INR, and (5) a set of learnable base parameters for the INR, which will be modulated by the output of each linear head $\texttt{Head}_k$. This improves training compared to directly producing the weights (Ortiz et al., 2023). The main difference between our framework and Trans-INR (Chen & Wang, 2022) is that our framework uses a pre-trained Transformer-based foundation model, whereas Trans-INR uses a Transformer encoder trained from scratch. Given an input data instance, such as a 2D view in the example above, it is discretized and embedded into $\mathbb{R}^d$ by the embedding layer $\texttt{Embed}$ to get data tokens $[t_1^0, \ldots, t_m^0]$. Superscripts indicate the number of attention blocks $B_i$ a token has passed through. For simplicity, unlike Trans-INR we do not use $\texttt{Embed}$ to tokenize task- or modality-specific auxiliary data such as camera pose information in the novel view synthesis task. The extra weight tokens are then concatenated to the data tokens and fed through the transformer encoder:

$$[t_1^i, \ldots, t_m^i, w_1^i, \ldots, w_q^i] = B_i([t_1^{i-1}, \ldots, t_m^{i-1}, w_1^{i-1}, \ldots, w_q^{i-1}]), 1 \le i \le N \qquad (1)$$

Finally, only the output tokens corresponding to weight tokens are used to generate the weights of each layer:

$$L_k = \texttt{Norm}(\texttt{Head}_k([w_{a_1}^N, \ldots, w_{a_r}^N]) * \texttt{BaseParam}_k) \qquad (2)$$

where $\texttt{Norm}$ is an operation that normalizes the weights to have unit $L_2$ norm. For computational efficiency, we keep the weight grouping scheme of Trans-INR, where each token only helps to generate the weights of a single layer, with the number of tokens $r$ being the total number of parameters in the layer divided by some hyperparameter $g$. More details can be found in Chen & Wang (2022). The model is trained end-to-end by generating the weights of the INR using the encoder ($\texttt{Embed}$ and $\texttt{Enc}$), using the INR to predict the data instance (see the previous section and Figure 1), and supervising the training with a reconstruction loss (not shown in Figure 1).

**Prompt Tuning Transformer-based Hypernetworks**   As our framework is based on the Transformer architecture, parameter-efficient fine-tuning (PEFT) methods for Transformers such as prompt tuning (Jia et al., 2022) can be used almost directly. Prompt tuning is particularly simple because our framework already has learnable prompt tokens in the form of the weight tokens (the red tokens in Figure 1), so (shallow) prompt tuning can be achieved by just freezing the weights of the pre-trained foundation model encoder, consisting of the embedding layer $\texttt{Embed}$ and Transformer encoder $\texttt{Enc}$, and fine-tuning the remaining weights, which consist of the learnable weight tokens $w_j^0$, INR weight-producing linear heads $\texttt{Head}_K$, and base parameters $\texttt{BaseParam}_k$. Unlike prompt tuning, the token input to the linear heads corresponds to the learnable prompt tokens and not the data tokens.

## 3 EXPERIMENTS

### 3.1 EXPERIMENTAL SETUP

**Pre-trained Backbones** We experiment with the following large pre-trained models: supervised ViT (Dosovitskiy et al., 2020) trained on ImageNet-21k (Deng et al., 2009), DeiT (Touvron et al., 2021), a supervised model trained on ImageNet-1k (Deng et al., 2009) using distillation, DINO (Caron et al., 2021), a self-supervised model pre-trained with self-distillation, DINO v2 (Oquab et al., 2023), which improves on DINO with additional curated training data and other improvements, CLIP (Radford et al., 2021), a large vision-language model trained using an image-text contrastive loss, and MAE (He et al., 2022), which is pre-trained with masked image modeling. For audio, we use Whisper (Radford et al., 2023), an encoder-decoder model that is self-supervised on a variety of speech tasks. For the Whisper model, we only use the encoder.

**Baselines** In addition to our base framework, which is based on Trans-INR (Chen & Wang, 2022), we also examine two state-of-the-art extensions which are easily adapted to our framework by replacing their Transform backbone with pre-trained foundation models. PONP (Gu et al., 2023), representative of neural process-based/probabilistic methods, improves on Trans-INR by adapting the neural process (NP) (Garnelo et al., 2018b) meta-learning algorithm for generalizable INR learning. PONP learns a probabilistic INR instead of the deterministic one used by Trans-INR, with the output layer of the INR producing mean and variance predictions instead of point predictions. Instead of using MSE as a reconstruction loss, PONP uses the maximum-likelihood loss of conditional NPs (Garnelo et al., 2018a). Instance Pattern Composers (IPC) (Kim et al., 2023), representative of weight-sharing methods, improves on Trans-INR by using low-rank weight modulation to modulate just one weight matrix of one layer of the INR to instance-specific parameters while sharing all other weights of the INR among all data instances.

**Training** Training is done in one of three settings: 1) *randomly initialized*, where all the weights of the model are randomly initialized and then trained, 2) *fine-tuned*, *foundation model* or *FM*, where the Transformer encoder in our framework is initialized with the weights of a foundation model and the whole model is fine-tuned, and 3) *frozen* or *prompt tuned*, which corresponds to our prompt tuning approach where the Transformer encoder is initialized with foundation model weights and frozen.

### 3.2 TASKS

We experiment on the following datasets and tasks:

**Novel view synthesis** We use the novel view synthesis (NVS) dataset of LearnIt (Tancik et al., 2021), which consists of 50 rendered views of shapes in the *cars*, *chairs*, and *lamps* categories of the ShapeNet (Chang et al., 2015) 3D object dataset. Given a set of views of an object and a new viewing direction, the objective is to generate a view that best matches the ground truth view in that viewing direction. To fairly compare among different pre-trained models, unless otherwise stated we only examine models using the ViT-B/16 architecture. We restrict to the case where a single input view of the object is given. Unlike previous works (Tancik et al., 2021; Chen & Wang, 2022; Guo et al., 2023; Gu et al., 2023; Kim et al., 2023), unless otherwise stated we train on all categories at once instead of training a separate model for each category, a much harder task, and also evaluate using the average performance on each category. We numerically evaluate all methods with four metrics that cover different aspects of image similarity: peak signal-to-noise ratio (PSNR), SSIM (Wang et al., 2004), LPIPS (Zhang et al., 2018b), and FID (Heusel et al., 2017).

**Audio reconstruction** For audio reconstruction, following IPC (Kim et al., 2023) we use the LibriSpeech-clean audio dataset (Panayotov et al., 2015). The framework is trained on randomly cropped audio, while test audio is trimmed to 1s for evaluation (Kim et al., 2023), which is done with PSNR. For this method, we only benchmark the IPC algorithm.

### 3.3 MAIN RESULTS

Our main results can be found in Tables 1, 2, 3. We find that:

Table 1: Comparison of different foundation models using the ViT-B/16 architecture as backbones on the NVS task. All backbones examined outperform random initalization except for MAE (He et al., 2022), which we hypothesize is due to the lack of global image representation learning.

| Backbone | PSNR ($\uparrow$) | SSIM ($\uparrow$) | LPIPS ($\downarrow$) | FID ($\downarrow$) |
|---|---|---|---|---|
| Randomly initialized | 20.862 | 0.8357 | 0.1511 | 0.2751 |
| MAE (He et al., 2022) | 20.701 | 0.8312 | 0.1753 | 0.2866 |
| Supervised (Dosovitskiy et al., 2020) | 21.324 | 0.8501 | 0.0966 | 0.1516 |
| DeiT (Touvron et al., 2021) | 21.587 | 0.8530 | 0.1125 | 0.1860 |
| DINO (Caron et al., 2021) | 21.737 | 0.8555 | 0.1101 | 0.1810 |
| CLIP (Radford et al., 2021) | 21.770 | 0.8556 | 0.1126 | 0.1834 |
| DINOv2 (Oquab et al., 2023) | 22.095 | 0.8609 | 0.1063 | 0.1705 |

1. **In general, hypernetworks with large pre-trained models as backbones outperform hypernetworks with the same architecture trained from scratch, but the choice of pre-trained model matters** (§3.4). Initializing hypernetwork weights from large pre-trained model improves performance in general, although not all foundation models lead to improvements due to differences in pre-training strategy. In particular, learning a good global image representation seems to be crucial.

2. **Foundation models improve generalization to classes unseen during training** (§3.5), but full fine-tuning may cause some forgetting of generalizable foundation model features.

3. **Hypernetworks with frozen foundation model backbones have at least comparable performance to hypernetworks with the same architecture trained from scratch** (§3.7), while using significantly fewer learnable parameters (100K vs 87M parameters).

4. **Foundation model-based hypernetworks scale** (§3.6, §3.8) Hypernetworks augmented with foundation models are both more data efficient (§3.6) and perform better with larger foundation models (§3.8).

5. **The effects of foundation models are robust over different algorithms and different modalities** (§3.9) We find that hypernetwork performance increases across different algorithms, including as neural process-based/probabilistic methods (Gu et al., 2023) and weight-sharing methods (Kim et al., 2023), as well as over different modalities (3D objects and audio).

## 3.4 FOUNDATION MODELS INCREASE HYPERNETWORK PERFORMANCE

Table 1 shows that fine-tuning our foundation model framework for hypernetworks outperforms training from a random initialization for almost all of the investigated foundation models, with the exception of MAE (He et al., 2022). We hypothesize that the poor performance of MAE is due to its masked image modeling self-supervision, which learns good mid-level interaction between image patches (Li et al., 2022a), but fails to learn good global features Liang et al. (2022). We find that the three best foundation models are CLIP (Radford et al., 2021), DINO (Caron et al., 2021), and DINOv2 (Oquab et al., 2023), which we hypothesize is due to these methods learning strong global image representations during pre-training. The contrastive pre-training objective of CLIP promotes the learning of global image representations (Li et al., 2024), whereas the DINO self-distillation objective (Caron et al., 2021) encourages the `[CLS]` token of both DINO and DINOv2 to learn a global image representation (Li et al., 2024). We hypothesize that learning good global image representations is crucial for the NVS and generalizable INR tasks.

Qualitative results can be found in Figure 4. We find that the foundation model approach is better at learning the shape of objects, such as the curved back of a chair.

## 3.5 FOUNDATION MODELS IMPROVE HYPERNETWORK GENERALIZABILITY TO UNSEEN CLASSES

Due to their large pre-training datasets, we hypothesize that fine-tuning from foundation model features should also lead to better zero-shot generalization to unseen classes. To test this, we train on

Table 2: Comparison of hypernetwork generalizability to classes unseen during training using random initialization, fine-tuning from DINOv2 (Oquab et al., 2023), and prompt tuning with frozen DINOv2. Each method was trained with only two of the classes in the ShapeNet NVS dataset and evaluated on the third, unseen class. The best metrics are highlighted in **bold**.

| Method | Training → Test | PSNR (↑) | SSIM (↑) | LPIPS (↓) | FID (↓) |
|---|---|---|---|---|---|
| Randomly initialized | cars, chairs → lamps | 17.377 | 0.7959 | **0.1898** | 0.1179 |
| Frozen | cars, chairs → lamps | **18.346** | 0.7972 | 0.2941 | 0.3033 |
| Fine-tuned | cars, chairs → lamps | 17.474 | **0.7977** | 0.1903 | **0.0956** |
| Randomly initialized | cars, lamps → chairs | 13.163 | 0.6212 | **0.3751** | 1.0199 |
| Frozen | cars, lamps → chairs | **13.536** | 0.6112 | 0.4279 | 1.0017 |
| Fine-tuned | cars, lamps → chairs | 13.322 | **0.6238** | 0.3845 | **0.9680** |
| Randomly initialized | chairs, lamps → cars | 15.431 | 0.7521 | 0.2987 | 0.3276 |
| Frozen | chairs, lamps → cars | **15.503** | 0.7465 | 0.3607 | 0.3623 |
| Fine-tuned | chairs, lamps → cars | 15.382 | **0.7692** | **0.2310** | **0.1548** |

two of the three classes in the ShapeNet NVS dataset and evaluate on the third class, which is unseen during training, using the *random* and *fine-tune* strategies. Furthermore, to see if full fine-tuning is degrading the generalizability of foundation model features through catastrophic forgetting, we train additional models where the foundation model is frozen. In Table 2, we find that the full fine-tuning of foundation models improves zero-shot generalization to unseen classes over training from scratch, and that reconstructions (as measured by PSNR) can be improved even further if the foundation model is frozen instead of fine-tuned, indicating that some of the generalizability of the pre-trained features is lost during full fine-tuning.

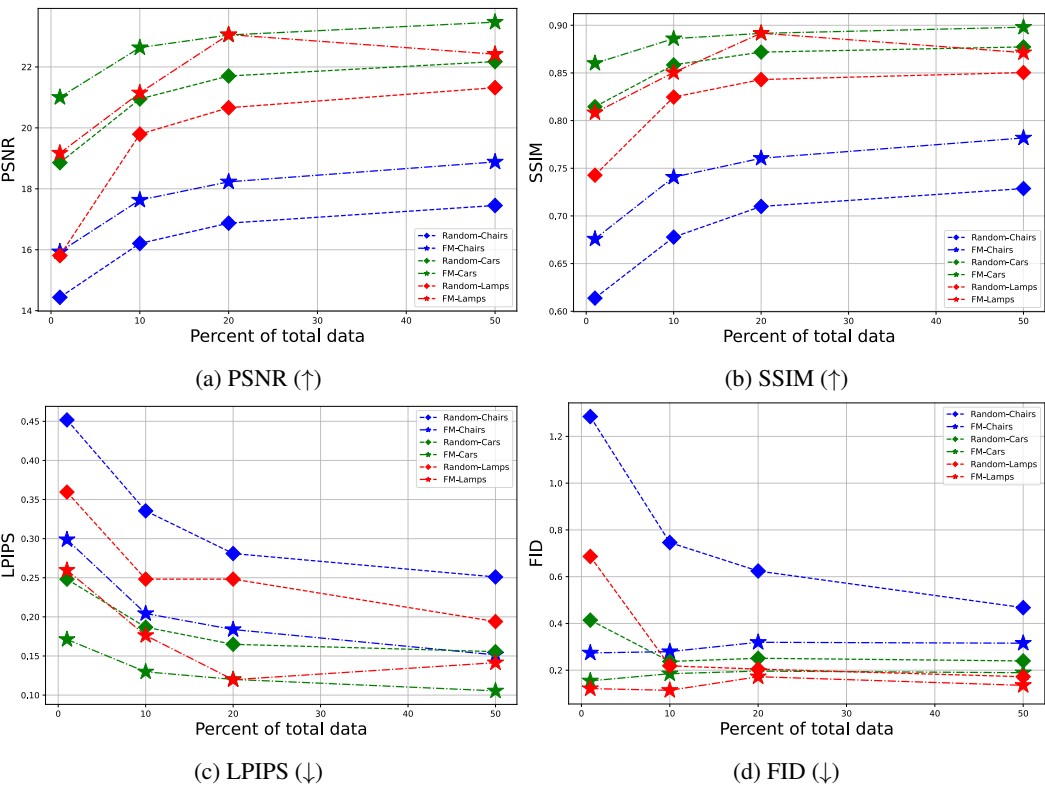

Figure 2: Plots showing performance vs the amount of training data for both the randomly initialized (Random) and foundation model (FM) strategies.

Table 3: Comparison of the three different training strategies using DINO (Caron et al., 2021) on the NVS task. We find that models with a frozen DINO encoder perform better than the same model randomly initialized on PSNR, while remaining close on the other metrics with a fraction of the parameters. Full fine-tuning results in a significant increase in performance for all metrics.

| Method | Trainable Parameters | PSNR ($\uparrow$) | SSIM ($\uparrow$) | LPIPS ($\downarrow$) | FID ($\downarrow$) |
|---|---|---|---|---|---|
| Randomly initialized | 87.3M (100%) | 20.862 | 0.8357 | 0.1511 | 0.2751 |
| Frozen | 100K (0.11%) | 21.035 | 0.8335 | 0.1767 | 0.3212 |
| Fine-tuned | 87.3M (100%) | 21.737 | 0.8555 | 0.1101 | 0.1810 |

Table 4: Comparison of the effectiveness of foundation models for different hypernetwork algorithms on the novel view synthesis task. We find that regardless of the algorithm type, using a foundation model significantly improves performance. The best performing models are **bolded**.

| Method | PSNR ($\uparrow$) | SSIM ($\uparrow$) | LPIPS ($\downarrow$) | FID ($\downarrow$) |
|---|---|---|---|---|
| Trans-INR (Chen & Wang, 2022) | 20.850 | 0.8346 | 0.1586 | 0.2680 |
| Fine-tuned Trans-INR | **22.095** | **0.8609** | **0.1063** | **0.1705** |
| PONP (Gu et al., 2023) | 20.878 | 0.8357 | 0.1584 | 0.2795 |
| Fine-tuned PONP | **21.993** | **0.8591** | **0.1083** | **0.1765** |
| Instance Pattern Composers (Kim et al., 2023) | 20.102 | **0.8344** | 0.1811 | 0.2655 |
| Fine-tuned Instance Pattern Composers | **20.672** | 0.8324 | **0.1450** | **0.1971** |

### 3.6 FOUNDATION MODELS IMPROVE HYPERNETWORK DATA EFFICIENCY

In Figure 2, we compare the random initialization and foundation model strategies when trained on 1%, 10%, 20%, and 50% of the data. We find that for PSNR, SSIM, and LPIPS, the foundation model approach significantly outperforms random initialization on every category, and that these metrics are closely correlated. We observe that, unlike the other metrics, FID seems to plateau quickly and may even increase slightly with more data. One possible explanation is that FID may not detect gradual improvements in image quality and may instead incorrectly indicate quality degradation (Jayasumana et al., 2024), which may be happening here as the image quality gradually improves due to the increasing amount of training data. This indicates that foundation model-based hypernetworks will be better able to leverage the increasingly larger datasets being published, such as Objaverse (Deitke et al., 2023).

### 3.7 FROZEN FOUNDATION MODELS ENABLE PARAMETER EFFICIENT HYPERNETWORKS

In Table 3, we find that even if the Transformer encoder weights are frozen, the model's performance can perform on par or even exceed that of the same model randomly initialized, despite using only a fraction of the learnable parameters (100K vs 87M). This means that even if there are no computational considerations, training a hypernetwork using the simple formula of extra input tokens, a frozen pre-trained backbone, and MLP heads is a promising approach. Surprisingly, despite the modality difference between image features and the weights of an INR, prompt tuning can succeed with only a linear head producing the weights of each layer.

### 3.8 SCALING LAWS FOR HYPERNETWORKS

As shown in Figure 3, we find that increasing the number of parameters of the foundation model generally increases performance on all metrics. This suggests that being able to scale foundation models to more parameters would directly lead to an increase hypernetwork performance. All foundation models investigated showed improved performance with more data with the exception of the supervised ViT (Dosovitskiy et al., 2020), where PSNR increased but all other metrics decreased, with the caveat that many models, including the supervised ViT, only had two model sizes tested. It has been observed before that for ViTs trained on image classification, better upstream performance does not necessarily result in better performance on downstream tasks (Zhai et al., 2022).

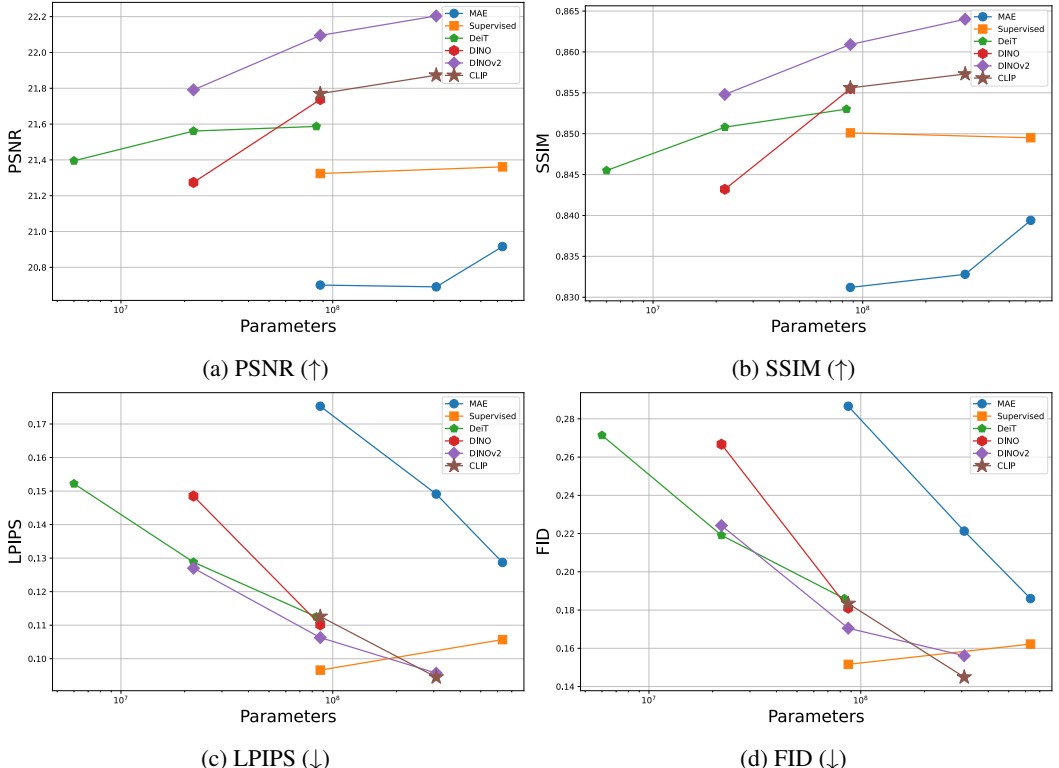

(a) PSNR (↑)                    (b) SSIM (↑)

(c) LPIPS (↓)                   (d) FID (↓)

Figure 3: Plots of NVS performance vs number of Transformer encoder parameters, as measured by the four metrics, on the NVS task using the Trans-INR algorithm. We find that increasing model size generally leads to increased performance, with supervised ViTs (Dosovitskiy et al., 2020) being a clear outlier.

## 3.9 ROBUSTNESS BETWEEN ALGORITHMS AND MODALITIES

Since our hypernetwork framework is based on Trans-INR (Chen & Wang, 2022), follow-up improvements (see Sec. 3.1) to this framework can also be enhanced with foundation models. In Table 4, we find that the improvement provided by using foundation model backbones persists regardless of the type of algorithm used (e.g. probabilistic (Gu et al., 2023) or weight-sharing (Kim et al., 2023)). We note that while past work showed that weight-sharing approaches (Kim et al., 2023; Lee et al., 2024) were state-of-the-art when training a separate model per category, they perform much worse than competing algorithms when training is done across categories. We hypothesize that this is due to these methods sharing too many of the INR parameters among all data instances, limiting the expressivity of the model and resulting in underfitting. This drop in performance holds with the addition of foundation models. In contrast, PONP continues to perform slightly better than Trans-INR in this setting, but with the addition of foundation models, fine-tuned Trans-INR performs slightly better than PONP.

Table 5 shows that this effect extends across different modalities to audio, indicating the robustness of the benefits of foundation models to hypernetworks. Notably, we see that parameter-efficient prompt tuning performs slightly better than a model with random initialization.

## 4 RELATED WORKS

**Implicit Neural Representations (INRs)**    INRs represent complex data such as 3D objects, scenes, and audio by parameterizing them using a neural network. Architectures for INRs include using Fourier features (Mildenhall et al., 2021; Tancik et al., 2020) and sinusoidal activation functions (Sitzmann et al., 2020b). The flexibility of the INR framework has led to applications in a wide variety of domains, including 3D shape and scene reconstruction (Mildenhall et al., 2021; Sitzmann

Table 5: Audio reconstruction on the LibriSpeech dataset using the weight-sharing hypernetwork approach of Instance Pattern Composers (Kim et al., 2023). The best performing model is **bolded**.

| Method | Params | PSNR (↑) |
|---|---|---|
| Randomly initialized | 72.7M (100%) | 24.431 |
| Prompt-tuned Whisper-base (Radford et al., 2023) | 100K (0.1%) | 24.432 |
| Fine-tuned Whisper-tiny (Radford et al., 2021) | 37.9M (52.1%) | 24.431 |
| Fine-tuned Whisper-base (Radford et al., 2023) | 72.7M (100%) | **24.434** |

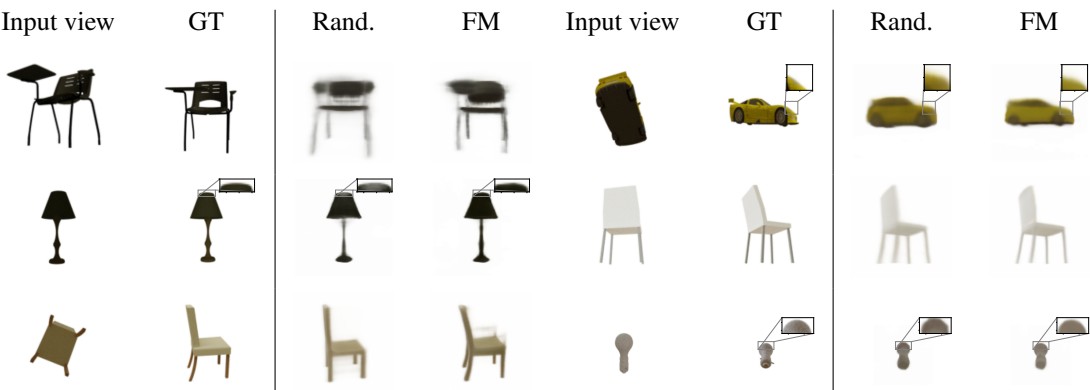

Figure 4: Comparison of qualitative results between the best foundation model-based hypernetwork and hypernetworks trained from scratch. Novel views generated with the hypernetwork approach (FM) are more faithful to the groundtruth than the baseline (Random). For example, the lamp in the middle row is better reconstructed at both the top of the lamp and on its stem, while for the two chairs the FM approach better captures their curved backs. You may need to zoom in to see the differences.

et al., 2019; 2020b;a), generative models (Poole et al., 2022; Liu et al., 2023), robotics (Li et al., 2022b), and more.

**Generalizable INR** The problem of learning a generalizable INR is usually formulated as a meta-learning task, where learning an INR for each signal is a separate task (Tancik et al., 2021; Chen & Wang, 2022; Gu et al., 2023). Early methods used auto-decoding (Mescheder et al., 2019; Park et al., 2019), where a latent vector is optimized per-instance and concatenated with the input to the INR. The current major approaches to this problem are gradient-based meta-learning, hypernetworks, and neural processes. The gradient-based meta-learning approach (Sitzmann et al., 2020a; Tancik et al., 2021; Lee et al., 2021) uses algorithms such as MAML (Finn et al., 2017) or Reptile (Nichol et al., 2018) to learn a good INR initialization that can be quickly finetuned, but has the disadvantage of requiring additional test-time optimization. Hypernetwork approaches (Mescheder et al., 2019; Sitzmann et al., 2020b; 2019; 2021; Chen & Wang, 2022) use a separate shared encoder that generates the weights of an INR, and have fast inference, as an INR can be generated in one forward pass of the encoder. Neural process approaches (Guo et al., 2023; Gu et al., 2023) use the neural process meta-learning framework (Garnelo et al., 2018b;a) which use neural networks to parameterize a stochastic process. This approach may be combined with hypernetwork approaches (Gu et al., 2023). Other approaches to generalizable INR follow the strategy of improving INRs by distillation from foundation models (Wang et al., 2022; Ye et al., 2023; Liao et al., 2024). In FeatureNeRF (Ye et al., 2023), foundation model features are distilled by training (non-hypernetwork) generalizable INRs to jointly predict foundation model features along with the reconstruction. Unlike these works, our model focuses only on improving hypernetwork architectures with foundation models.

**Hypernetworks** Hypernetworks are neural networks that produce or modify the parameters of another network. In this paper, we focus on hypernetworks that generate implicit neural representations. Hypernetworks are used as a means of conditioning implicit neural representation generation and also to create generalizable implicit neural representations. Early works using hypernetworks

to generate implicit neural representations (Mescheder et al., 2019; Sitzmann et al., 2020b; 2019; 2021). Early hypernetworks methods for generating INRs used simpler MLP (Sitzmann et al., 2019; 2021) or convolutional (Mescheder et al., 2019; Sitzmann et al., 2020b) architectures for the hyper-network. Trans-INR (Chen & Wang, 2022) proposed using the more powerful vision transformer (ViT) (Dosovitskiy et al., 2020) as the base architecture for hypernetworks, and this method has been improved upon by incorporating neural processes (Gu et al., 2023) or using weight modulations to learn only some of the layers of the INR while sharing the rest of the parameters (Kim et al., 2023; Lee et al., 2024). Our work examines the impact of foundation models on hypernetworks using Transformer-based architectures, which to the best of our knowledge has not been examined before.

## 5 Conclusion

We present a rigorous investigation of using foundation models to improve hypernetworks for generalizable INR tasks, providing key insights for designing future hypernetwork models. We demonstrate that foundation models improve hypernetwork performance on both seen and unseen classes, and show that this effect is robust. We also provide a parameter-efficient way to create hypernetwork models based on prompt tuning. We also further analyze the effect of using foundation models, looking at the choice of foundation as well as scaling with data and parameters. We hope that our investigation serves as a starting point for investigating foundation models for hypernetwork architectures.

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
