# OpenReview forum: "Foundation Models Secretly Understand Neural Network Weights: Enhancing Hypernetwork Architectures with Foundation Models"
_ICLR.cc/2025/Conference — ICLR 2025 Poster_

### Official Review · Reviewer_xDrF · 2024-10-28

**Soundness:** 3
**Presentation:** 4
**Contribution:** 3
**Rating:** 8
**Confidence:** 3

**Summary:**

This paper investigates the utility of foundation models as backbones for hypernetworks, that is, neural networks which generate the weights for INRs. The authors show that fine-tuning (or freezing and prompt-tuning) foundation models such as DINO and CLIP can indeed lead to hypernetworks that outperform the same architectures trained from scratch.

**Strengths:**

This paper investigates a logical idea, is clearly written and offers a very thorough analysis, in the sense that many questions one might have about the proposed approach are investigated. For example, the authors investigate different foundation models as backbones, compare different fine-tuning methods and hypernetwork algorithms, as well as different tasks. I can imagine that using foundation models as starting points for training INRs will become the standard procedure in certain visual domains.

**Weaknesses:**

* As somebody unfamiliar with the NVS task, it would have been nice to not only compare to the (potentially weak?) baseline of training from random initialisation, but also to see the performance of the current state-of-the-art method, as a point of reference. I can see that using the FM backbone is better than the baseline, but is it really good in absolute terms? Looking into Tancik et al 2021, it seems like the average PSNR of their best method is 21.333, while that of Chen and Wang 2022 achieves 22.07 on average. I realize that the numbers are not directly comparable, but why did you choose to train one model on all three tasks instead of training individual models?
* I wonder about the variance of the performance values in Table 1: Ideally, one would train each model multiple times and give an estimate of the standard deviation of performance. I’m not yet convinced that the model differences are really stable. The same holds for table 2.
* Finally, I would have liked to see an explicit discussion of limitations and shortcomings of the method.

**Questions:**

## Questions
* Is there really enough data in NVS to justify the training of models with 86M parameters from scratch?

## Additional Feedback
* Figure 1 could match the text a bit better: In the figure, the linear layers labelled as “Positional Encoding” are called BaseParam in the text (I think? Or are these the heads?) and the Embed layer and Enc are not labelled in the figure. It is still understandable, but would remove any uncertainty to label the elements in the figure better.
* It would be good to motivate and differentiate the four metrics (PSNR, SSIM, LPIPS, FID).
* Line 141f (“For computational efficiency, each token only helps to generate the weights of a single layer, with the number of tokens r being the total number of parameters in the layer divided by some hyperparameter g.”) is unclear to me, is BaseParam_k a sparse matrix?
* I think that “Enhancing Hypernetwork architectures with foundation models” would have been a fine title on its own, but that might just be preference (and will not affect my rating of the paper)

## Other thoughts
(going beyond what would be reasonable to do within the rebuttal period)

In addition to the three training settings you investigate (starting in line 180), have you considered training LoRA adapters as a middle ground between freezing the FM and fine-tuning it completely? Maybe this could help with the issue of catastrophic forgetting mentioned in line 230.

## Nitpicks
* Line 323 "seems to plateau"
* Line 407 "decreased"
* Line 416 "is persists"

---

> ### Author Response · Authors · 2024-11-22
>
> Thank you for the time and effort you put into reading and reviewing our paper and for pointing out that our paper “offers a very thorough analysis” and that our idea of "using foundation models as starting points for INRs" may become the standard.
>
> Addressing your comments and questions:
>
> Re: including SOTA references
>
> Thanks for pointing this out! We would like to note that like you mentioned, the best foundation model-based hypernetwork in Table 1 outperforms LearnIt [1] and Trans-INR [2], in a more difficult setting (training one model on all three classes instead of having a separate model for each class). The main reason we choose to train one model on all three classes instead of training individual models is to reduce the total number of models we would need to train.
>
> Re: variance of models
>
> We agree that having multiple runs and providing an estimate of the standard deviation is valuable. Our past experiments with this architecture indicate that the variance of multiple runs is low (in the range of 0.03 - 0.13 PSNR). We are currently running multiple runs with DINOv2 to verify this. Since there are many models evaluated in our paper, we don’t have time to perform multiple runs on all the models.
>
> Re: limitations
>
> One limitation of our method is not tokenizing task-specific information such as pose and camera parameters for novel view synthesis, as it’s not clear how to incorporate this with a pre-trained ViT tokenizer. Previous results suggest that this may further improve performance. Another limitation is that, to compare with past work, we have only used the simple volume renderer of [7], but better results could be obtained by using a more sophisticated volume renderer. Another limitation is that we only investigate fine-tuning and using a frozen foundation model, but approaches such as using LoRA fine-tuning, as you suggested, may perform even better. Another limitation is that we did not have time to investigate how scaling to large datasets such as Objaverse [3] affects performance. We will include this discussion of limitations in the updated paper.
>
> Re: enough NVS training data
>
> Although we were unable to investigate it in our paper, there are existing datasets that should be large enough. Examples include the Objaverse [3]-derived dataset (800K+ objects, 12 views per object) used to train Zero-1-to-3 [4] and the dataset used to train Large Reconstruction Model (LRM) [5], which combines Objaverse [3] and MVImgNet [6], consisting of 730K+ 3D objects with 32 rendered views per object and 220K+ videos of objects, respectively.
>
> Re: Figure 1
>
> Thanks for pointing this out! To clarify, the positional encoding was meant to label $\gamma(v)$, and the base parameters are not shown in our current figure. We will make the figure more clear by moving the labels and including the base parameters into the figure.
>
> Re: metrics
>
> We will add a discussion of the motivations and differences between the different metrics to the paper.
>
> Re: BaseParams_k
>
> To clarify, there is no restriction on the rank of BaseParam_k; it is an arbitrary learnable dense matrix. Line 141F was intended to describe the weight grouping scheme of [2], which we also use in our method; we will make this clear by referencing [2].
>
> Re: title
>
> We appreciate the suggestion! While we think that the first part of the title conveys one of our contributions, if you feel strongly that the first part of the title is unnecessary, we will change it.
>
> Re: LoRA
>
> Thank you for this suggestion! This seems like a very interesting direction for future work.
>
> Minor: Thanks for pointing out these mistakes in the writing. We will correct them in the revised version.
>
> [1] Tancik, Matthew, et al. "Learned initializations for optimizing coordinate-based neural representations." Proceedings of the IEEE/CVF Conference on Computer Vision and Pattern Recognition. 2021.
>
> [2] Chen, Yinbo, and Xiaolong Wang. "Transformers as meta-learners for implicit neural representations." European Conference on Computer Vision. Cham: Springer Nature Switzerland, 2022.
>
> [3] Deitke, Matt, et al. "Objaverse: A universe of annotated 3d objects." Proceedings of the IEEE/CVF Conference on Computer Vision and Pattern Recognition. 2023.
>
> [4] Liu, Ruoshi, et al. "Zero-1-to-3: Zero-shot one image to 3d object." Proceedings of the IEEE/CVF international conference on computer vision. 2023.
>
> [5] Hong, Yicong, et al. "Lrm: Large reconstruction model for single image to 3d." arXiv preprint arXiv:2311.04400 (2023).
>
> [6] Yu, Xianggang, et al. "Mvimgnet: A large-scale dataset of multi-view images." Proceedings of the IEEE/CVF conference on computer vision and pattern recognition. 2023.
>
> [7] Tancik, Matthew, et al. "Learned initializations for optimizing coordinate-based neural representations." Proceedings of the IEEE/CVF Conference on Computer Vision and Pattern Recognition. 2021.

---

> ### Author Response · Authors · 2024-11-25
> **Variance for Table 1**
>
> Here are the results of our experiment with 3 runs trained with the same parameters in the setting of Table 1:
>
> | Run | PSNR | SSIM | LPIPS | FID |
> |-----------|-----------|-----------|-----------|-----------|
> | Run #1 | 20.4678633 | 0.8213314476 | 0.2027442273 | 0.3511989017 |
> | Run #2 | 20.4529114 | 0.8211523911 | 0.205314859 | 0.3651582698 |
> | Run #3 | 20.51477549 | 0.8231690394 | 0.2013994583 | 0.3569750488 |
> | St. Dev. | 0.03227868218 | 0.001116219469 | 0.001989426781 | 0.007014187265 |
>
> Edit: for completeness, here are the resuls of 3 runs trained with the same parameters, in the cars, chairs $\to$ lamps setting of Table 2:
>
> | Run | PSNR | SSIM | LPIPS | FID |
> |-----------|-----------|-----------|-----------|-----------|
> | Run #1 | 18.42807124 | 0.8059116034 | 0.2476512101 | 0.2277166545 |
> | Run #2 | 18.31853344 | 0.8048780737 | 0.2426565284 | 0.2282271832 |
> | Run #3 | 18.41760341 | 0.804099671 | 0.2458745585 | 0.2283564806 |
> | St. Dev. | 0.0604468995 | 0.0009089548272 | 0.002531766638 | 0.0003383131524 |

---

> > ### Comment · Reviewer_xDrF · 2024-11-26
> >
> > Dear Authors, \
> > thank you for the response, and for running the experiment to verify the stability of results, which I find convincing. I maintain my score, as I still think it is surprising that using foundation model weights works better than training from scratch in this very special setting, and you explore this thoroughly, as far as I can tell (although I must also underline my lack of familiarity with this area). \
> > I do not feel strongly about the title, feel free to keep it as is.

---

> > > ### Author Response · Authors · 2024-11-27
> > >
> > > Thank you for helping us improve our paper! Your suggestion and feedback have been invaluable.

---

### Official Review · Reviewer_QLSE · 2024-11-03

**Soundness:** 4
**Presentation:** 3
**Contribution:** 2
**Rating:** 5
**Confidence:** 3

**Summary:**

There are many applications/tasks where large pre-trained (foundation) models can be potentially helpful, but the possibility has not been explored. The authors identify one of these being generalizable INRs using hypernetworks (hypernets).

The paper proposes to use transformer-based foundation models within an architecture based on Trans-INR. The authors show that the model can be trained either through full fine-tuning or through a prompt-tuning-based approach that keeps the foundation model encoder weights frozen. In addition to the input patches as input to the transformer, the authors add learnable weight tokens corresponding to the network weights to be generated at the output.

The authors show that hypernets with foundation model backbones show improved generalization to seen and unseen classes, and improved data and parameter efficiency, compared to hypernets trained from scratch. They note that foundation model choice is important, claiming that models that have learned a good global representation (as opposed to local representations like with MAE) perform better. The authors also show that the scaling benefits of larger foundation models transfer to the generalizable INR task through their incorporation in hypernets. Finally, the authors demonstrate that their idea is robust across two other hypernet algorithms based on Trans-INR and between two modalities (vision and audio).

**Strengths:**

- The idea is simple and new to the specific task of generalizable INRs using hypernets.
- The authors conduct an extensive empirical evaluation to support their claims and show that the gains are reproduced across algorithms and modalities.
- Most of the paper is easy to follow.

**Weaknesses:**

Primary:
- The technical contribution of this paper is replacing a transformer component that has previously been trained from scratch with a pre-trained foundation model, which is not original and is now a fairly standard recipe. Please see the questions section for a possible actionable avenue.

Secondary:
- It is confusing that in Table 2, random initialization outperforms foundation models on LPIPS in 2/3 settings, but fine-tuning performs much better on the remaining setting. The paper does not provide an explanation.
- It is not clear what "positional encoding" refers to in Figure 1 and the related discussion. In the context of transformers, this term denotes adding position information to the transformer inputs, which is not the case here. It is also hard to see if the linear heads underneath are labeled positional encodings or the $\gamma(\mathbf{v})$ block to its right. Most likely, this refers to the MLP's output, but there is some ambiguity.
- There is no discussion of training hyperparameters used for the experiments. Are all of these the same as what the base frameworks used?

Minor (no impact on score):
- It is unclear what type of norm is considered around line 140 (context: "normalizes the weights to have norm 1"). It is likely L2-norm based on Trans-INR, but it should be specified here.
- Section 3.1: In the first line, "we experiments" -> "we experiment".
- Around line 175: "Instead using of MSE" -> "Instead using MSE".
- Last sentence in the abstract is too long (the entire point 2).

**Questions:**

Have you considered deviations from the Trans-INR architecture? Are there any components that become unnecessary or even detrimental with the addition of foundation models (as a motivating example that is not directly related, strong regularization that helps at smaller scales can limit performance gains at larger scales)?

---

> ### Author Response · Authors · 2024-11-22
>
> Thank you for the time and effort you put into reading and reviewing our paper and for pointing out that our idea is “simple and new”.
>
> Addressing your comments and questions:
>
> Reviewer comment: The technical contribution of this paper is replacing a transformer component that has previously been trained from scratch with a pre-trained foundation model, which is not original and is now a fairly standard recipe. Please see the questions section for a possible actionable avenue.
>
> We would like to point out that one of our contributions is to show that this approach can work in the hypernetwork setting, which is a much different setting than the typical setting where pre-trained transformers are used. In our case, the pre-trained encoder is expected to modify the weights of a neural network, with only an extra linear head to transform its output. We show that even when the pre-trained encoder is frozen, we can still achieve state-of-the-art results, despite the apparent modality gap between neural network weights and image representations and while only using a fraction of the parameters. We argue that due to the modality difference between neural network weights and foundation model representations, it was not clear that this approach would have worked.
>
> Reviewer question: Have you considered deviations from the Trans-INR architecture? Are there any components that become unnecessary or even detrimental with the addition of foundation models (as a motivating example that is not directly related, strong regularization that helps at smaller scales can limit performance gains at larger scales)?
>
> The main components are the ViT encoder, INR decoder, extra learnable tokens, linear heads, and base parameters. The extra learnable weight tokens are necessary to ensure there are enough tokens to modify all the parameters of the INR, and the linear heads are necessary to convert the tokens output by the ViT encoder to the correct shape for modifying the INR’s weight matrices. The base parameters are so we can use a residual learning approach; without them, the model doesn’t converge. Although this is not in our current paper, we have explored improvements to the architecture, such as increasing the number of layers in the head. Surprisingly, in our experiment, this did not increase performance, though further tuning is probably needed. With regards to deviations from the Trans-INR architecture, we note that unlike Trans-INR, we do not tokenize task-specific side information (e.g., pose information or camera parameters). We leave this out to be able to use each foundation model’s pre-trained tokenizer. We will clarify and add this to the paper.
>
> Reviewer comment: It is confusing that in Table 2, random initialization outperforms foundation models on LPIPS in 2/3 settings, but fine-tuning performs much better on the remaining setting. The paper does not provide an explanation.
>
> Although random initialization is slightly better than foundation model fine-tuning in LPIPS in 2 out of 3 settings in Table 2, the difference is very small, and the difference in the third setting is much larger. Across all metrics, foundation model fine-tuning still outperforms training from a random initialization. We don’t know a clear reason why this is but this would be interesting to explore further.
>
> Reviewer comment: It is not clear what "positional encoding" refers to in Figure 1 and the related discussion. In the context of transformers, this term denotes adding position information to the transformer inputs, which is not the case here. It is also hard to see if the linear heads underneath are labeled positional encodings or the γ(v) block to its right. Most likely, this refers to the MLP's output, but there is some ambiguity.
>
> Thanks for pointing this out! The positional encoding label refers to the part labeled $\gamma(v)$. We will move the $\gamma(v)$ label to make this more clear. We will also add the architecture of the INR to make this more clear in the text of the paper.
>
> Reviewer comment: There is no discussion of training hyperparameters used for the experiments. Are all of these the same as what the base frameworks used?
>
> For most of the experiments, we use the same hyperparameters as the base frameworks. For the novel view synthesis task, these hyperparameters are 1000 training epochs, a learning rate of 1e-4 with a decay to 1e-5 at 800 epochs, and a batch size of 128. For the audio reconstruction task, these hyperparameters are 100 training epochs, learning rate 1e-4, and a batch size of 64. We will provide the full details in the updated paper.
>
> Minor: Thanks for pointing out these mistakes in the writing. We will correct them in the revised version.

---

> > ### Comment · Reviewer_QLSE · 2024-11-26
> >
> > Thank you for your responses. I have read through them, and have decided to maintain my rating.

---

> ### Author Response · Authors · 2024-11-26
>
> Thank you for considering our responses to your comments.
>
> We would like to add additional discussion re: deviations from the Trans-INR architecture. The algorithms in Table 4 (PONP [1], IPC [2]) may also be considered deviations from the Trans-INR architecture. As described in the paper, PONP, inspired by neural processes, deviates from Trans-INR by replacing the point predictions of the INR with predictions of the mean and standard deviation of a Gaussian distribution at each coordinate. Since the predictions are now distributions, PONP is trained with a maximum likelihood loss from the conditional neural process literature. Instance Pattern Composers (IPC) deviates from Trans-INR by only modifying one of the layers of the INR with the hypernetwork per signal, and sharing the rest of the parameters of the INR among all signals. In the case where one model is trained per class, past work shows that IPC is the best performing algorithm with PONP second, but in our case, where one model is trained for all three classes, IPC is the worst performing algorithm, which we hypothesize is due to the oversharing of INR parameters among instances of different classes. This drop in performance holds with the addition of foundation models. In contrast, PONP continues to perform slightly better than Trans-INR in this setting, but with the addition of foundation models, fine-tuned Trans-INR performs slightly better than PONP. We will make this more clear in the paper.
>
> [1] Gu, Jeffrey, Kuan-Chieh Wang, and Serena Yeung. "Generalizable Neural Fields as Partially Observed Neural Processes." Proceedings of the IEEE/CVF International Conference on Computer Vision. 2023.
>
> [2] Kim, Chiheon, et al. "Generalizable implicit neural representations via instance pattern composers." Proceedings of the IEEE/CVF Conference on Computer Vision and Pattern Recognition. 2023.

---

### Official Review · Reviewer_JGgX · 2024-11-04

**Soundness:** 3
**Presentation:** 3
**Contribution:** 3
**Rating:** 6
**Confidence:** 4

**Summary:**

The paper proposes to leverage pretrained foundation models for enhancing hypernetworks which are trained to generate weights of implicit neural networks, i.e. networks whose parameters are a representation of a single data sample.  The approach leverages transformer based architectures to learn a (modulated) set of weights vectors that will correspond to the implicit representation of each sample. The method is evaluated on mainly on image data, with an additional experiment on audio data, respectively in a novel image synthesis task and audio reconstruction. The experimental section tests varies aspects and benefits of incorporating foundation models from performance, to sample efficiency, generalization to unseen class and parameter efficiency.

**Strengths:**

- The idea is simple and effective. Leveraging information stored in pretend foudnation models for enhancing implicit neural representations networks is mostly a novel idea to the best of my knowledge and it may benefits future research in the area.

- The experimental section is quite broad  and comprehensive: it shows the benefits of incorporating foundation models hyper networks to learn implicit representations from the point of view of:  performance, sample complexity, parameter efficiency and multi modality. This seems to suggest that in general pretrained vision models should be incorporated  in learning INRs and pave the way to new applications in this direction.

- In particular, the results by just finetuning the heads is quite impressive, pointing at the fact that the connection between weights and features should be further inspected in future works.

- The paper writing is overall clear.

**Weaknesses:**

- The architecture and training procedure of the model although it builds on previous work could be explained better in the paper, for See related questions in the question section

- On the experiment on audio data the model doesn't seem to benefit much form the foundation model. Do the authors have an intuition of why? Could it be related with the complexity of the task or to the fact that audio FM are less expressive in general than Vision ones?

- It would be interesting to see some results on a different task/ dataset on image data.

- The following work should be included and discussed in the related works section, as it also explores the relation between features extracted and performance of implicit neural networks:

     - Ye, J., Wang, N., & Wang, X. (2023). Featurenerf: Learning generalizable nerfs by distilling foundation models. In Proceedings of the IEEE/CVF International Conference on Computer Vision (pp. 8962-8973

Similarly, yet fare as line of works the following works which distill information in CLIP embeddings to learn better Implicit representations should be discussed:

- Wang, Can, et al. "Clip-nerf: Text-and-image driven manipulation of neural radiance fields." Proceedings of the IEEE/CVF Conference on Computer Vision and Pattern Recognition. 2022

- Liao, Guibiao, et al. "Ov-nerf: Open-vocabulary neural radiance fields with vision and language foundation models for 3d semantic understanding." arXiv,  2024



*Minor*

- Visualization quality of Figure 4 could be improved, for example by putting box to zoom on details.

- I spotted the following typos:

    - Figure 3 caption: "leads generally leads" -> "generally leads"
    - Figure 4 caption: "baselin" -> "baseline"
    - Line 174: "using of" -> "of using"
    - Lin e 408: "descreased" -> "decreased"

**Questions:**

- Integration of FM into existing methods :
    -  How is the architecture and training strategy related to the method proposed? Is the model still trained with meta-learning?
    -  For the experiments in Table 4, again how is the FM included in the methods specifically? Is there any adaptation to the architecture or training strategy?

- Do the authors have an intuition on why the FID metric seems to grow for foundation models based models in Figure 2?

---

> ### Author Response · Authors · 2024-11-22
>
> Thank you for the time and effort you put into reading and reviewing our paper and for pointing out that our method is "simple and effective" and our “results by just finetuning the heads is quite impressive”.
>
> Addressing your comments and questions:
>
> Re: reviewer weakness 1 and Q1, on architecture
>
> For the first question, we are looking to improve existing hypernetwork architectures with foundation models, so the architecture and training strategy described is largely that of previous works (e.g. [1]). The main difference is that we replace the transformer encoder with pre-trained ViT-based vision foundation models, with the rest of the components being the same as [1] except that we do not tokenize task-specific information (more below).
>
> For the second question, hypernetworks are a type of meta-learning algorithm that can be trained almost exactly like a normal neural network. First, the input is passed through the hypernetwork, which instantiates the weights of an (empty) INR. This INR is then used to reconstruct the input, and a reconstruction loss is applied. Finally, the loss is backpropagated to the weights of the hypernetwork. The INR itself has no learnable weights; its weights are entirely provided by the hypernetwork. The only extra step compared to normal neural network training is the extra step where the hypernetwork instantiates the weights of the INR. During inference, we can just feed the input to the hypernetwork and get an INR as output, allowing us to generate a good INR in just one forward pass of the network. This approach is related to gradient-based meta-learning methods such as MAML (see [1]).
>
> Regarding Table 4, the three listed algorithms [1-3] are all variants of the transformer hypernetwork originally proposed in [1], so in all three cases foundation models could be incorporated the same way, by replacing the transformer encoder and tokenizer with the corresponding parts of a ViT-based foundation model. The main difference between [1] and [2] is the number of instance-specific weights modified by the hypernetwork. The main difference between [1] and [3] is that [3] outputs a distribution and thus has a maximum likelihood reconstruction loss.
>
> For the last question, to keep the comparison fair we do not tokenize task-specific side information (such as poses and camera parameters for novel-view synthesis), as is done in [1-3]. No other adaptations to the architecture and training strategy were used.
>
> We will clarify all these points in the updated version of the paper.
>
> Re: reviewer weakness 2, on audio:
>
> We conjecture that the small difference in performance is probably due to the low complexity of the task. Experiments on the more difficult 3s audio reconstruction task [2] show similar behavior.
>
> Re: reviewer weakness 3, on different tasks:
>
> We agree that results on different tasks would be interesting. However, at the time this paper was written, we were not aware of any transformer-based hypernetwork architectures for other image tasks such as image classification, segmentation, or depth prediction, and developing new hypernetwork approaches is out of the scope of this paper. A related discussion is in the response to reviewer s6Rw.
>
> Re: reviewer weakness 4, on related works:
>
> Thank you for pointing out these related works. We will include a discussion of these works in the updated version of our paper. Although it is not the goal of our paper to compare hypernetworks to other approaches, we are running an additional experiment to compare FeatureNeRF to our method as requested by reviewer s6Rw on novel view synthesis and will update when we have results.
>
> Re: Question 2, on FID:
>
> While FID is a popular and well-established metric, it has drawbacks. One of these drawbacks is that FID may not detect gradual improvements in image quality and may instead incorrectly indicate quality degradation [4], which may be happening here as the image quality gradually improves due to the increasing amount of training data. We will add this discussion to the paper.
>
> Minor: Thanks for suggesting the box zoom and pointing out the typos. We will fix these in the revised paper.
>
> [1] Chen, Yinbo, and Xiaolong Wang. "Transformers as meta-learners for implicit neural representations." European Conference on Computer Vision. Cham: Springer Nature Switzerland, 2022.
>
> [2] Kim, Chiheon, et al. "Generalizable implicit neural representations via instance pattern composers." Proceedings of the IEEE/CVF Conference on Computer Vision and Pattern Recognition. 2023.
>
> [3] Gu, Jeffrey, Kuan-Chieh Wang, and Serena Yeung. "Generalizable Neural Fields as Partially Observed Neural Processes." Proceedings of the IEEE/CVF International Conference on Computer Vision. 2023.
>
> [4] Jayasumana, Sadeep, et al. "Rethinking fid: Towards a better evaluation metric for image generation." Proceedings of the IEEE/CVF Conference on Computer Vision and Pattern Recognition. 2024.

---

> > ### Author Response · Authors · 2024-11-27
> >
> > Results for our experiment comparing our foundation-model enhanced hypernetwork to FeatureNeRF can be found in the comments under Reviewer s6Rw's review.

---

> ### Comment · Reviewer_JGgX · 2024-11-27
> **Response to rebuttal**
>
> I thank the author for the answers and the additional experiments conducted in the rebuttal period.
>
> I'm satisfied by the answers and I appreciated the experimental comparison with FeatureNeRf, therefore I stand by my recommendation of acceptance.

---

> > ### Author Response · Authors · 2024-12-03
> >
> > Thank you for your time and effort! Your suggestions and feedback have greatly improved our paper.

---

### Official Review · Reviewer_s6Rw · 2024-11-10

**Soundness:** 2
**Presentation:** 2
**Contribution:** 2
**Rating:** 3
**Confidence:** 4

**Summary:**

The authors propose a novel method of using visual foundation models as part of a hyper-network to prediction the weights of an MLP to perform a task. The authors evaluate their method on a novel viewpoint reconstruction task and an audio reconstruction tasks using three methods: random initialization, fine-tuning, and promp-tuning. They demonstrate a small performance improvement using a foundation model and then fine-tuning or prompt-tuning over random initialization.

**Strengths:**

- The authors have selected and interesting question to investigate and have clearly describe their approach (aside from lacking INR model details)
- This work demonstrates a performance improvement of using foundation model weights over a random baseline.
- The overall architecture proposed is simple and generalizable to any foundation model or task.

**Weaknesses:**

- The overall performance improvement is small.
- The authors chose tasks that are very difficult to evaluate and only two tasks were evaluated.
- The paper lacks details about the exact architecture and scale of the INR network which seems like an import parameter that would be interesting to vary.
- The evaluation lacks other baseline methods of training the INR network such as distillation. Although not the goal of this paper, evaluation of other training methods seems important for contextualizing the performance of this method.

**Questions:**

- What is the exact architecture and number of parameters in the INR network?

- Why did the authors chose generative tasks where the available evaluation metrics such as PSNR, SSIM, LPIPS, and FID are a poor proxy measurement for model performance on the task? Why not use image classification, segmentation, depth prediction, or many other tasks that are easier to evaluate and have more model baselines to compare against?

- How does the performance achieved by the predicted INR weights from this method compare to distilling a foundation model fine-tuned for these tasks into the INR network?

---

> ### Author Response · Authors · 2024-11-22
>
> Thank you for the time and effort you put into reading and reviewing our paper and for pointing out that we have "selected an interesting question to investigate" and that our method is "simple and generalizable".
>
> Addressing your comments and questions:
>
> Reviewer comment: The overall performance improvement is small.
>
> The performance difference between the best of our proposed foundation model-enhanced hypernetworks and the best baseline is larger than the performance difference between previously published comparable work in the field and their baselines. For example, on the same task [1] shows a 0.733 gain in PSNR over the best baseline on the same task, [2] shows only a 0.253 gain in PSNR, [3] shows only a 0.077 gain in PSNR, and while [4] does not publish their exact numbers on this task, but estimating from Figure 4(a) of their paper their gain in PSNR is around 0.3. For the same task, we can show a gain of 1.233 in PSNR (Table 1 of our paper) with the best foundation model backbone.
>
> Reviewer comment: The authors chose tasks that are very difficult to evaluate and only two tasks were evaluated.
>
> We agree that evaluating generated novel views is a difficult task. However, we would like to point out that we are using the commonly accepted metrics for this task, including PSNR [1-7], SSIM [5, 7], LPIPS [5, 6], FID [5, 6], and we would also like to emphasize that we have provided a variety of different metrics to make our evaluation more robust. Previous hypernetwork-based works for this task [1-4] only used PSNR as a metric.
>
> In terms of our choice of task, please see the below response to the reviewer question “why did the authors choose generative tasks…”
>
> Reviewer comment: The paper lacks details about the exact architecture and scale of the INR network which seems like an important parameter that would be interesting to vary.
>
> Reviewer question: What is the exact architecture and number of parameters in the INR network?
>
> As in previous work [1-4], our INR architecture is an MLP with 6 layers of hidden dimension 256, positional encoding, and ReLU activations. We will add this information to the paper.
>
> With respect to varying the parameters of the INR, see the below response to the reviewer question on distillation.
>
> Reviewer comment: The evaluation lacks other baseline methods of training the INR network such as distillation. Although not the goal of this paper, evaluation of other training methods seems important for contextualizing the performance of this method.
>
> We agree that although this is not the goal of our paper, adding a comparison to distillation from foundation models will help provide context to our results. We are running an additional experiment comparing our method to FeatureNeRF [7], a distillation approach also mentioned by reviewer JGgX. Since FeatureNeRF reports novel-view synthesis results using a different INR architecture, this experiment can also be used to compare versions of our method with different INR architectures. We will report the results when available.
>
> Reviewer question: Why did the authors choose generative tasks where the available evaluation metrics such as PSNR, SSIM, LPIPS, and FID are a poor proxy measurement for model performance on the task? Why not use image classification, segmentation, depth prediction, or many other tasks that are easier to evaluate and have more model baselines to compare against?
>
> We chose these tasks because hypernetworks are commonly used for these tasks, and in this setting hypernetworks have the advantage of providing more expressive conditioning of INRs, do not have to rely on a single vector for conditioning or grid-based representations, and can mimic gradient-based meta-learning, another technique for conditioning INRs, without requiring the computation of higher-order derivatives [1].
>
> Although we agree evaluation can be challenging in this domain, the metrics we use are well-established by many papers (see above) and we believe that they are sufficient to demonstrate the utility of our approach.
>
> At the time this paper was written, we were not aware of any transformer-based hypernetwork architectures for image classification, segmentation, or depth prediction. We have since found [8], which proposes a hypernetwork approach for segmentation using a U-Net within a U-Net architecture and an EfficientNet backbone for the hypernetwork, which lacks pre-trained foundation models. Using this architecture would require special adaptation, which is out of the scope of this work.

---

> > ### Author Response · Authors · 2024-11-22
> > **Citations**
> >
> > [1] Chen, Yinbo, and Xiaolong Wang. "Transformers as meta-learners for implicit neural representations." European Conference on Computer Vision. Cham: Springer Nature Switzerland, 2022.
> >
> > [2] Kim, Chiheon, et al. "Generalizable implicit neural representations via instance pattern composers." Proceedings of the IEEE/CVF Conference on Computer Vision and Pattern Recognition. 2023.
> >
> > [3] Gu, Jeffrey, Kuan-Chieh Wang, and Serena Yeung. "Generalizable Neural Fields as Partially Observed Neural Processes." Proceedings of the IEEE/CVF International Conference on Computer Vision. 2023.
> >
> > [4] Lee, Doyup, et al. "Locality-aware generalizable implicit neural representation." Advances in Neural Information Processing Systems 36 (2024).
> >
> > [5] Liu, Ruoshi, et al. "Zero-1-to-3: Zero-shot one image to 3d object." Proceedings of the IEEE/CVF international conference on computer vision. 2023.
> >
> > [6] Hong, Yicong, et al. "Lrm: Large reconstruction model for single image to 3d." arXiv preprint arXiv:2311.04400 (2023).
> >
> > [7] Ye, Jianglong, Naiyan Wang, and Xiaolong Wang. "Featurenerf: Learning generalizable nerfs by distilling foundation models." Proceedings of the IEEE/CVF International Conference on Computer Vision. 2023.
> >
> > [8] Nirkin, Yuval, Lior Wolf, and Tal Hassner. "Hyperseg: Patch-wise hypernetwork for real-time semantic segmentation." Proceedings of the IEEE/CVF conference on computer vision and pattern recognition. 2021.

---

> ### Author Response · Authors · 2024-11-27
> **Comparison against distillation**
>
> Although it is not the goal of our paper, as you requested we compare the performance of our foundation model-enhanced hypernetwork method to FeatureNeRF [1], a method that distills foundation model features into INRs. Although FeatureNeRF is not designed for the novel-view synthesis task, performing only comparably to the baseline reported in their paper, we compare our method to FeatureNeRF as it is the only foundation model distillation method that we are aware of which provides experimental results on the novel view synthesis task.
>
> For our experiment, we train FeatureNeRF for single-view novel view synthesis using the LearnIt ShapeNet dataset [2] used in our paper. Although this dataset and FeatureNeRF's dataset are both rendered from ShapeNet, the LearnIt dataset is more difficult because only 25 reference views are provided (compared to 50 for FeatureNeRF) and the camera may be under the object. We investigate FeatureNeRF's perfomance in two settings: one where a separate model is trained for each of the three classes (cars, chairs, and lamps), which is the setting in the FeatureNeRF paper. We also evluate FeatureNeRF on the harder task of using one model for all three classes. For our method we train it only in the latter setting, as in our paper. To make the comparison fair, we train our model with an INR architecture of 5 layers of hidden dimension 512, which is essentially the INR architecture used by FeatureNeRF. For reference, the INR architecture for the experiments in our paper has 6 layers of hidden dimension 256. For this experiment, we use the [official code](https://github.com/jianglongye/featurenerf) for FeatureNeRF and their default hyperparameters in the provided DINO config.
>
> Visually, we find that FeatureNeRF can only learn a blob that is roughly car-, chair-, or lamp-like,  and generally fails to recover the shape and the color of the image.  Surprisingly, despite the visual appearance of the novel views, the PSNR is high, highlighting the limitations of PSNR as a metric for this task. All other metrics are significantly lower than that of our model, reflecting the visual appearance of the novel views and highlighting the robustness of evaluating the task with a variety of complementary metrics. Interestingly, we find that if we instead train and evaluate FeatureNeRF while rendering a white background instead of the default black background, performance on all metrics is significantly degraded, while visually the rendered objects look at least comparable. For example, PSNR goes from $\sim 21.1$ to $\sim 2.5$ in the case of class-specific models. Our model is trained with the renderer rendering a white background.
>
> For our method, we provide comparisons to both the prompt-tuned (frozen) version and the fine-tuned version using both DINOv2 and DINO. Our model performs slightly worse in this setting than with the default INR architecture in our paper, but still greatly outperforms FeatureNeRF overall in both settings. We will provide results for the DINO backbone when available.
>
> | Model | PSNR ($\uparrow$) | SSIM ($\uparrow$) | LPIPS ($\downarrow$) | FID ($\downarrow$) |
> |-----------|-----------|-----------|-----------|-----------|
> | FeatureNeRF (class-specific, black bg) | 21.171 | 0.215 | 0.355 | 0.695 |
> | FeatureNeRF (combined, black bg) | 20.460 | 0.069 | 0.361 | 1.226 |
> | Ours (DINO prompt-tuned) | 20.554 | 0.819  | 0.218 | 0.395 |
> | Ours (DINO fine-tuned) | 21.774 | 0.855 | 0.115 | 0.198 |
> | Ours (DINOv2 prompt-tuned) | 20.504| 0.819 | 0.218 | 0.368 |
> | Ours (DINOv2 fine-tuned) | 21.691 | 0.851 | 0.134 | 0.173 |
>
> Edit: updated the table with DINO results. We see that DINO performs comparably to DINOv2 and greatly outperforms FeatureNeRF, and may be a more direct comparison because FeatureNeRF distills DINO features.
>
> [1] Ye, Jianglong, Naiyan Wang, and Xiaolong Wang. "Featurenerf: Learning generalizable nerfs by distilling foundation models." Proceedings of the IEEE/CVF International Conference on Computer Vision. 2023.
>
> [2] Tancik, Matthew, et al. "Learned initializations for optimizing coordinate-based neural representations." Proceedings of the IEEE/CVF Conference on Computer Vision and Pattern Recognition. 2021.

---

### Author Response · Authors · 2024-11-26
**Revised paper**

Thank you for your thoughtful reviews and constructive feedback on our manuscript. We appreciate your feedback and have made significant revisions to improve the paper. In particular, we have added new supplementary material including training details (Reviewer QLSE), INR architecture details (Reviewer s6Rw), discussion of metrics (Reviewer xDrF), a comparison to previous results (Reviewer xDrF), and limitations (Reviewer xDrF). In the main paper, we have updated Figure 1 (Reviewers QLSE, xDrF), added zoom to Figure 4 (Reviewer JGgX), and clarified and fixed errors in the writing as discussed in the reviews (all reviewers). Your reviews have improved the quality of our writing significantly. If there is any additional feedback on our manuscript, we would appreciate it.

---

### Meta-Review · Area_Chair_NoU4 · 2024-12-23

**Metareview:**

Reviewers are in overall agreement that the proposed method is simple, effective and novel. The manuscript contains a comprehensive evaluation and is clearly written. During the rebuttal, the authors provided a thorough explanation and rebuttal for a variety of questions, based on which I would encourage the authors to update the manuscript. As many of the criticisms are easily addressed (e.g. providing more details on experimental settings and architectural parameters), I would recommend acceptance.

**Additional Comments On Reviewer Discussion:**

While the majority of the reviewer discussion was positive, I was concerned with the lack of response/update provided by reviewer s6Rw despite several reminders. As the authors ran additional experiments and a full explanation/rebuttal of several of the criticisms, I would have expected a positive reviewer response / potential score increase, which would have been particularly significant, due to their relatively low score. As a result, I have chosen to down weight this reviewer's assessment.

---

### Decision · Program_Chairs · 2025-01-22

Accept (Poster)